# Cows Get Crohn’s Disease and They’re Giving Us Diabetes

**DOI:** 10.3390/microorganisms7100466

**Published:** 2019-10-17

**Authors:** Coad Thomas Dow, Leonardo A Sechi

**Affiliations:** 1McPherson Eye Research Institute, University of Wisconsin, 9431 WIMR, 1111 Highland Avenue, Madison, WI 53705, USA; 2Department of Biomedical Sciences, University of Sassari, Viale San Pietro 43b, 07100 Sassari, Italy

**Keywords:** *Mycobacterium avium* ss. *paratuberculosis*, MAP, Crohn’s, Johne’s, zoonosis, CARD15, SLC11a1, type 1 diabetes mellitus, T1D, molecular mimicry, ZnT8, HSP65, HERV, TRIGR study, sarcoidosis, Blau syndrome, multiple sclerosis, Hashimoto’s thyroiditis, lupus, Parkinson’s disease, rheumatoid arthritis

## Abstract

Increasingly, Johne’s disease of ruminants and human Crohn’s disease are regarded as the same infectious disease: paratuberculosis. *Mycobacterium avium* ss. *paratuberculosis* (MAP) is the cause of Johne’s and is the most commonly linked infectious cause of Crohn’s disease. Humans are broadly exposed to MAP in dairy products and in the environment. MAP has been found within granulomas such as Crohn’s disease and can stimulate autoantibodies in diseases such as type 1 diabetes (T1D) and Hashimoto’s thyroiditis. Moreover, beyond Crohn’s and T1D, MAP is increasingly associated with a host of autoimmune diseases. This article suggests near equivalency between paucibacillary Johne’s disease of ruminant animals and human Crohn’s disease and implicates MAP zoonosis beyond Crohn’s disease to include T1D.

## 1. Introduction

*Mycobacterium avium* ss. *paratuberculosis* (MAP) is the cause of paratuberculosis, or Johne’s disease. It is mostly studied in ruminant animals such as cattle, goats and sheep. After one hundred years of controversy, there is a warming to the notion that MAP is also the zoonotic cause of the similar Crohn’s disease of humans [1,2,3,4]. Koch’s postulates are the criteria used to establish a causal relationship between microbe and disease. These postulates state that the microbe must: (1) be found in all cases of the disease, (2) be recovered and maintained in pure culture, (3) be capable of producing the original infection even after several generations in culture and (4) be retrievable from an inoculated animal and cultured again. The basis of the hundred-year controversy is the fact that traditional culturing (and staining) has been largely unsuccessful in identifying MAP in human samples [2,3,4]. Some have argued that these criteria have been met, tying MAP to Crohn’s disease [1,2]. Others contend that the postulates were established for acute infectious diseases and do not equally apply to chronic diseases like paratuberculosis, wherein individuals may become infected but remain in a latent subclinical state without developing a clinical disease, despite a positive culture and/or PCR [5,6,7,8,9,10]. Such latency is also seen in tuberculosis, where the estimated ratio of healthy infected carriers to new TB patients is 219:1 [11]. MAP is very difficult to culture from humans and eludes detection. MAP can exist with a modified cell wall—the component of the bacterium that takes up the characteristic acid stain. MAP can shed its cell wall, becoming a spheroplast or L-form (Figure 1) [12]. The bacterium is then no longer “acid fast” and cannot be detected microscopically in the traditional manner. This morphologic change allows MAP to become spore-like. The spore morphotype capable of surviving heat and other stressors enables MAP to persist in host macrophages and in the environment [13]. Adding to the difficulty of microscopic confirmation of MAP is that MAP, as with leprosy [14] and tuberculosis [15], can persist in a paucibacillary form (low numbers of observed organisms) [10]. Culture-independent methods such as PCR offer a more rapid indication of the presence of MAP than culture [16,17].

In 2004, Naser was able to culture MAP from the blood of Crohn’s patients [18]. The article was published in *The Lancet* and was featured on the cover. It read: “We detected viable *Mycobacterium avium* subspecies *paratuberculosis* in peripheral blood in a substantial proportion of individuals with Crohn’s disease, adding to the evidence for a role of the organism in the aetiology of this disease.”

This report “resulted in vigorous debate in the literature.” The authors were challenged to reproduce the study in a blind multi-center investigation. They did. Samples were split between four labs: three dedicated labs for MAP and a medical reference lab. All the labs were able to grow MAP except the medical reference lab [19]. This is at the heart of the century-long controversy—it is difficult to detect MAP with older laboratory methods. In 2005, Sechi and associates, in the largest series to date, reported the isolation of MAP from intestinal mucosal biopsies of Crohn’s patients [20]. Of note, MAP has been cultured from the breast milk of patients with Crohn’s disease [21,22]. The linkage of Crohn’s and Johne’s, with contemporary methods, has been validated in testing tissue at both a cellular and molecular level [23].

MAP-associated diseases have been explored due to the identification of shared genetic risks for the specific disease and concomitant mycobacterial infection. Investigations of polymorphisms of the CARD15 (NOD2) [2,24,25], SLC11a1 (NRAMP1) [26,27,28], LRRK2 [29,30], PTPN2/22 [31] and VDR [32] genes have proven fruitful as they impart a permissive state for mycobacterial infection due to the disruption of pathogen recognition and/or phagosome maturation. These genes have been linked to MAP and the following diseases: Crohn’s disease [2,28], Blau syndrome [2], multiple sclerosis [2,33], autoimmune (Hashimoto’s) thyroiditis [34,35,36], Parkinson’s disease [29,37], rheumatoid arthritis [27,31,38], lupus [39] and T1D [32,40].

## 2. MAP and Human Exposure

According to the USDA (United States Department of Agriculture), the herd-level prevalence of MAP infection in US dairy herds has increased from 21.6% in 1996 to 91.1% in 2007 [41]. MAP is present in pasteurized milk [42,43], infant formula made from pasteurized milk [44], surface water [45,46,47,48], soil [45], cow manure ‘‘lagoons’’ that can leach into surface water, cow manure in both solid and liquid forms that is applied as fertilizer to agricultural land [49,50] and municipal tap water [51]. All of these provide multiple opportunities for MAP exposure. In an Ohio study of domestic tap water, DNA of MAP was found in over 80% of the samples [52]. Filtration and chlorination, normal water treatment processes, kill off MAP competitors, thereby amplifying mycobacteria organisms instead of killing them [53]. Moreover, mycobacteria organisms grow on plastic water bottles [54], on tap water pipes [55] and in biofilms [56]. A study testing 65 samples of infant formula from 18 countries found more than 40% of samples were positive for viable MAP [57].

## 3. MAP and Crohn’s

Increasingly, Johne’s and Crohn’s are being regarded as the same infectious disease: paratuberculosis [58,59,60,61]. Studies show detection and isolation of MAP in adults with chronic Crohn’s [20], as well as with newly diagnosed pediatric Crohn’s patients [62]. Meta-analyses have shown that a majority of studies associating MAP with Crohn’s demonstrate MAP infection in the Crohn’s patients [63,64]. MAP has been cultured from peripheral blood white cells in a range from 50% to 100% of patients with Crohn’s disease, and less frequently from healthy individuals [65].

As noted, paratuberculosis is a global disease. Extensive testing in India describes an increasing MAP “bio-load” in cattle (43%), buffalo (36%), goats (23%) and sheep (41%). Moreover, in this same geographic area, 30.8% of 28,291 humans (via serum ELISA, blood PCR and stool PCR) tested positive for MAP [61]. Particularly telling of MAP as a zoonotic agent is the resolution of Crohn’s disease with anti-mycobacterial drugs targeting MAP [60]. The combination of clarithromycin, rifabutin and clofazimine, all anti-mycobacterial drugs, has shown efficacy as a primary treatment for Crohn’s disease [66].

An ongoing clinical trial is the RHB-104 study. The treatment is a proprietary combination of these same three anti-mycobacterial drugs for individuals with moderately severe or severe Crohn’s disease. The RHB-104 study is a multi-center, international interventional open clinical trial with 331 participants. Its primary outcome: “Reduction of the total Crohn’s Disease Activity Index (CDAI) score to less than 150” [67]. Results of the RHB-104 study were reported at the 2018 United European Gastroenterology meeting. Adding the anti-MAP therapy to standard therapy provided a “clinically meaningful and statistically significant treatment effect…” The trial data are from the published abstract LB06 [68]. Of note, RHB-104, the same three anti-mycobacterial drugs, is in clinical trial for another MAP-related disease, multiple sclerosis [69].

## 4. MAP and Diabetes

T1D occurs with the autoimmune destruction of the insulin-producing cells of the pancreas. It is most often seen in childhood or adolescence [70]. T1D has been on the increase since the last half of the 20th century [71]. Autoantibodies to pancreatic protein glutamic acid decarboxylase (GAD) are found in newly diagnosed children with T1D. These autoantibodies are felt to be the result of molecular mimicry whereby a foreign antigen (introduced by a bacterium) provokes an immune response, which then cross-reacts with a similar host protein [72,73,74]. The GAD pancreatic enzyme shares the sequence and conformational structure with the mycobacterial heat shock protein 65 (HSP65) [75]. In one study, all newly diagnosed T1D children had an immune response to mycobacterial HSP65 [76]. Multiple studies have associated T1D with exposure to cow’s milk [77,78,79]. Lastly, the TRIGR study reported no association, but they did not search for MAP presence [80].

In 2006, Dow postulated that MAP may be an environmental trigger for T1D in the genetically at-risk. Three proposals were offered to support the postulate: (1) there are shared genetic susceptibilities to both mycobacterial infection and T1D, (2) MAP is the source of the HSP65 protein, providing epitope homologies between mycobacterial HSP65 and pancreatic glutamic acid decarboxylase (GAD) and (3) epidemiologic findings tie the risk of T1D to early life exposure to cow’s milk [32].

Subsequently, Sechi and associates conducted several studies associating MAP and T1D. They found an association of MAP and T1D patients on their home island of Sardinia [81,82,83]. The island of Sardinia has the second highest incidence of T1D in the world [84]. They reported finding MAP in T1D patients but not in type 2 diabetics [85,86]. They found MAP in T1D children [87,88,89]. They confirmed a genetic risk factor linking mycobacterial infection and T1D [40]. They also identified additional MAP peptides that are homologous with pancreatic proteins [83,90,91] and showed that immune reaction to these MAP peptides cross-react to the classical islet cell antibodies [92]. They demonstrated parallel findings on the Italian mainland [93,94].

Recently, a body of evidence pointed to a role for human endogenous retroviruses (HERVs) in the activation of genes [95]. It is thought that most HERVs are genetically silent. However, assorted environmental stimuli, including infection, may activate HERVs to potentiate certain autoimmune diseases [96]. A recent study demonstrated anti-HERV antibodies correlating with sero-reactivity against MAP in children at risk for T1D [97]. This study showed that an activated HERV gene expressing a specific envelope protein, HERV-W, is associated with T1D in three distinct populations.

Of more than a dozen articles implicating MAP in T1D, only one article failed to do so. That article came from India where MAP was not found in the blood of T1D patients. A few possible explanations offered included the compulsory BCG vaccination against tuberculosis, with the thought that BCG provides cross protection against paratuberculosis as it does with leprosy. Also, the cultural culinary practice of vegetarianism would reduce exposure to MAP, as would the common practice of boiling milk before consumption [98].

Of interest is the publication that the BCG vaccination of long standing T1D individuals, followed by a booster in six months, resulted in the control of blood sugar (seen after a delay of three years). The effect was durable with normal blood sugars eight years after the vaccination [99,100]. The beneficial effect is postulated to be due to a “reset” of the immune system. An alternative explanation is that BCG vaccination is effective against MAP [101].

## 5. Discussion

The etiopathology of MAP-related human disease is multifaceted. MAP can initiate a granuloma, stimulate autoantibodies via molecular mimicry and potentiate immune imbalance by activating other stimulants, as found in MAP’s relation with HERVs [95,97].

It is extraordinary that a single pathogen can so significantly impact global animal agriculture, food safety and human disease. This is likely due to MAP’s ability to cause chronic regional intestinal inflammation in so many species [59]. Despite calls to address MAP zoonosis [102,103,104], those calls have been largely unheeded. Moreover, anti-MAP humoral reactivity was recently correlated between serum lipoprotein levels in subjects at T1DM risk (rT1DM) grouped by geographical background and in patients affected by MS or RA [105].

Interestingly, a review of the therapeutic agents for Crohn’s inflammation suggested that previously prescribed anti-inflammatory agents were actually treating MAP [106,107,108], while the strong contemporary anti-inflammatory TNF inhibitors are permissive for both MAP and tuberculosis [109,110,111].

Sufficient evidence points to the fact that until MAP is eliminated from the food chain, it may continue to be said that cows get Crohn’s disease and they are giving us diabetes, multiple sclerosis, sarcoidosis, Blau syndrome, Hashimoto’s thyroiditis, lupus, Parkinson’s disease and rheumatoid arthritis.

## Figures and Tables

**Figure 1 microorganisms-07-00466-f001:**
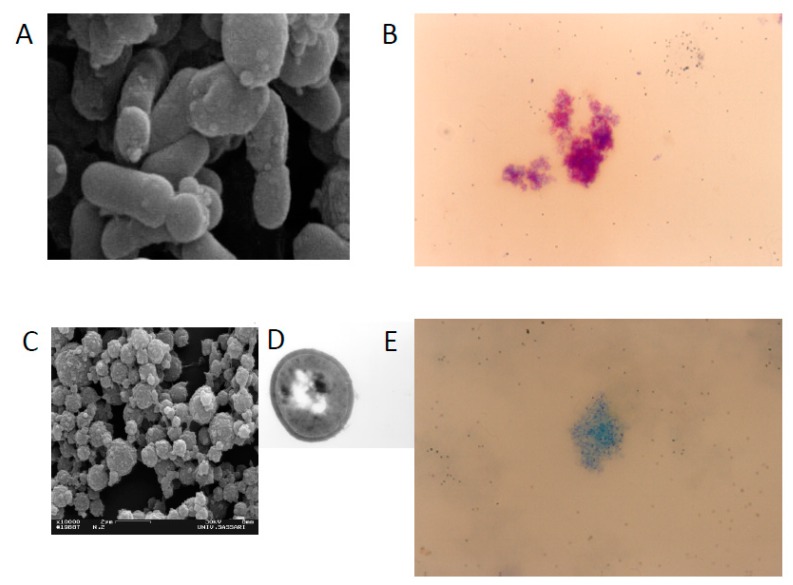
Scanning electron microscopy and Ziehl Neelsen staining of MAP cells IS900 positives growth in absence of Lysozime (**A**,**B**, respectively) with a bacillary shape and wild type cell wall. Scanning, Transmission electron microscopy and Ziehl Neelsen staining of MAP cells IS900 positives growth in presence of Lysozime when the bacteria lost the cell wall that takes up the characteristic acid stain (**C**–**E** respectively) with a round shape and cell wall deficient form.

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
