# Peer review of "Cows Get Crohn’s Disease and They’re Giving Us Diabetes"

_microorganisms, 2019, doi:10.3390/microorganisms7100466_

Round 1

Reviewer 1 Report

The author presents an hypothesis linking exposure to MAP via Johne's disease dairy cows through consumption of milk to the potential development of T1D. This is a significant claim and while this article is a review of literature- better development of the hypothesis would benefit the publication well. The following are recommendations toward improving the readability of the review.

Abstract:

Johns's disease in ruminants and Crohn's disease in humans is a complex disease with many triggers - both environmental and genetic. MAP is not the only causative link to IBD. This is not clear in the paper nor the abstract; and the authors opinion needs to be better substantiated, or clarified that MAP is one of many putatuve causes of Crohn's disease.

page3:

 Punctuation  ".... MAP is the cause of paratuberculosis, or John's disease. It is mostly studied..." in ruminants such as cattle..."

2nd paragraph is contentious without further development of the idea- my suggestion is to re organise paragraphs 2 and 3 and combine them such that the reasons why Koch's postulates are not met in this case are described earlier. However- while the postulates cannot be met by culturing techniques, the author presents literature demonstrating  MAP presence via PCR techniques. It would be better to spend less energy on reasons why they are not met and further describe why the PCR technique was successful despite the paucibacillary form of Mycobacterium species and how this now fulfills the postulates.

page 4:

The main argument in the abstract is that John's and Crohn's are equivalent- yet there is only 1 reference (self cite) demonstrating a link of MAP to Crohn's. As one of the main arguments- I suggest the author expand this link deeper. The finding of MAP in breast milk of Crohn's sufferers as a causal link is also stretched as only 2 pateints were positive. It is also not certain whether MAP causes Crohn's in susceptible individuals or selectively colonises mucosa but not initiate or perpetuate inflammation. Both the pros and cons of the equivalency would be beneficial here.

Page 5:

2nd paragraph not clear. Remove underlines and replace with better explanation "... identification of shared genetic risks for the specific disease and concomittant mycobacterial infection".

In what way have these genes been linked to MAP (expand).

Plagiarism  to a small extent was detected p6 "MAP can be cultured from the peripheral mononucler cells from 50-100" although referenced- better to rephrase this claim. As this review paper is modeled closely to [McNees AL, Markesich D, Zayyani NR, Graham DY. Mycobacterium paratuberculosis as a cause of Crohn's disease. Expert Rev Gastroenterol Hepatol. 2015; 9(12):1523-34. ] ; it may be better to stay well clear of copying entire sentences from this publication.

page 6: " ... resolution of Crohn's disease with antibimycobacterial drugs..." reference 58 is a review paper- please find the original publication and present the number of patients treatment was effective for as I believe numbers demonstrating success is still quite low.

please combine paragraphs 2,3,4 on page 7

page8 ".... The GAD pancreatic enzyme shares sequence and conformational..."

"subsequently, Sechi and associates ..."

page 9 : 2nd paragraph "... HERVs are genetically silenced (or silent ???)"

What is the link between active HERV protein and MAP in at risk children ? please expand on this idea.

"Of interest.... due to a "reset" of the immune system...." Other publications speak to a lack of effectiveness of tuberculosis drugs to MAP clearance- why / how does BCG vaccination work... expand. 

reference 99 is a news report- find the original publication. [Kuhtreier WM, Tran L, Kim T, Dybala M, Nguyen B, et al. (2018) Long-term reduction in hyperglycemia in advanced type 1 diabetes: The value of induced aerobic glycolysis with BCG vaccinations. npj Vaccines 3: 1-14.]

page 10 2nd paragraph " It is extroadinary"... this sentence is unclear rewrite .

Last paragraph looks messy- rewrite. Sufficient evidence points to the fact that  "until MAP..." amongst other pathologies of the modern world ... remove "....AND:"

Author Response

Dear Reviewer

Thank you for the valuable review of our manuscript.

Abstract
We modified with your suggested verbiage.

Page 3

We cleaned up the sentence punctuation.

We considered removing the Koch’s postulate portion of the discussion; but, in the end left it in with a re-write.

Page 4

We’ve combined sentences added references and deemphasized the breast milk reference.

Page 5

We’ve removed the underline removed the bridging sentence and incorporated your suggested language.  With this change, the answer to your question In what way have these genes been linked to MAP (expand), briefly, it is already in the manuscript: “disruption of pathogen recognition and/or phagosome maturation”

The plagiarism claim is taken seriously – reworded.

Page 6

The next sentence is referenced by #64:

            Chamberlin W, Borody T, Campbell J. Primary treatment of Crohn’s disease: combined antibiotics taking center stage. Expert Rev Clin Immunol. 2011;7:751–760.

We hope that rather than reiterating the controversy of MAP/Crohn’s we can make our zoonosis case with RHB-104 results and, moreso, by bringing T1D into the discussion.

Page 7

We combined the 3 paragraphs.

Page 8

Made both suggested changes.

Page 9

Silent – yes, working too late at night.

We’ve added this to the HERV discussion:

            “This study showed that an activated HERV gene expressing a specific envelope protein, HERV-W, is associated with T1D in three distinct populations.”

Clearly, this is only the beginning of the HERV-autoimmune story,

- why / how does BCG vaccination work

I, CTDow, am preparing another manuscript for this special edition entitled

            Proposing BCG Vaccination for Mycobacterium avium paratuberculosis (MAP)

                                                   Associated Autoimmune Diseases

The article includes a quote from

            Netea MG, van Crevel R. BCG-induced protection: effects on innate immune memory. Semin    Immunol. 2014 Dec;26(6):512-7. doi: 10.1016/j.smim.2014.09.006. Epub 2014          Oct 23.

“…despite the epidemiological evidence for heterologous protective effects of BCG vaccination, the perceived lack of biological plausibility has been a major obstacle in recognizing and in investigating these effects.”

I intend to argue that the biological plausibility is that of TB and leprosy, these diseases have a mycobacterial cause and/or stimulus.

reference 99 is a news report- find the original publication. [Kuhtreier WM, Tran L, Kim T, Dybala M, Nguyen B, et al. (2018) Long-term reduction in hyperglycemia in advanced type 1 diabetes: The value of induced aerobic glycolysis with BCG vaccinations. npj Vaccines 3: 1-14.]

The above reference you cite is our reference #97, #99 is a letter to editor suggesting an alternative rationale for the success of BCG in T1D.

Page 10

Rewritten,

amongst other pathologies of the modern world

That autoimmune diseases are considered by some, to be “diseases of civilization” will be an emphasis of the B

Reviewer 2 Report

Sechi and Dow nicely present the importance of MAP in the context of the human diseases. I think the subject was very good reviewed and represented in a good way that everyone can the concept easily. I don't have any further suggestions to authors. However, I have seen that they mentioned about Figure 1 in the manuscript. But I couldn't see this figure anywhere in the file. I would suggest either to include or to alter the text accordingly.

Author Response

Dear Reviewer,

We are grateful to you about the comments on the manuscript. We included Figure 1 and apologize for the mistake.

Thank you indeed

Leonardo A Sechi